# Targeting Inflammation and Skin Aging via the Gut–Skin Axis: The Role of *Lactiplantibacillus plantarum* HY7714-Derived Extracellular Vesicles

**DOI:** 10.3390/microorganisms12122466

**Published:** 2024-11-30

**Authors:** Hayera Lee, Yun-Ha Lee, Dong-Ki Hong, Sung-Jun Mo, Soomin Jeon, Soo-Dong Park, Jae-Jung Shim, Jeong-Lyoul Lee, Jae-Hwan Lee

**Affiliations:** R&BD Center, hy Co., Ltd., 22, Giheungdanji-ro 24beon-gil, Giheung-gu, Yongin-si 17086, Republic of Korea; yera@hy.co.kr (H.L.); yunha1006@hy.co.kr (Y.-H.L.); dkhong@hy.co.kr (D.-K.H.); stal6000@hy.co.kr (S.-J.M.); 10003273@hy.co.kr (S.J.); jjshim@hy.co.kr (J.-J.S.); jlleesk@hy.co.kr (J.-L.L.); jaehwan@hy.co.kr (J.-H.L.)

**Keywords:** gut–skin axis, *Lactiplantibacillus plantarum*, extracellular vesicles, intestinal inflammation, anti-photoaging

## Abstract

Intestinal mucosal tissues are prone to infections, often leading to inflammation. Lactic acid bacteria in the gut can modulate these inflammatory responses, but the interaction between host cells and lactic acid bacteria remains unclear. This study examines how *Lactiplantibacillus plantarum* HY7714 alleviates intestinal inflammation using gut-on-a-chip technology and in vitro models. Inflammation was induced using a gut-on-a-chip, and changes in cell morphology and barrier function were analyzed. Extracellular vesicles (EVs) derived from HY7714-improved intestinal cell structure repaired damage and restored tight junction integrity. Additionally, they attenuated inflammatory cytokines by regulating the MyD88/mTOR/NF-κB signaling pathway. RNA sequencing revealed downregulation of vicinal oxygen chelate (VOC) family proteins and proline aminopeptidase, both linked to inflammation and extracellular matrix interactions in skin health. Therefore, we explored the effects of HY7714 EVs on skin cells. The findings showed that HY7714 EVs reduced cytotoxicity and downregulated metalloproteinase expression in skin cells exposed to UVB radiation, indicating their potential anti-aging and anti-photoaging properties. These findings suggest that HY7714-derived EVs enhance both intestinal and skin health by reducing inflammation and improving barrier function, with potential benefits for the gut–skin axis.

## 1. Introduction

The intestines and the skin are active immune system components constantly exposed to external environments, such as bacteria and viruses. These organs must maintain homeostasis with various symbiotic organisms. The human intestine is inhabited by 10^13^–10^14^ microorganisms, and the intestinal mucosal tissue is easily infected by nosocomial viruses and bacteria [1]. Infectious colitis is caused by various bacteria and viruses, including *Shigella* and *Escherichia coli* [2]. These invasive pathogens penetrate mucous membranes and disrupt the epithelial barrier and mucus, leading to the presence of red and white blood cells in the stool and causing an acute inflammatory response [2,3]. Intestinal mucosal tissues have a protective immune reaction that triggers a rapid and powerful immune response as a primary defense against nosocomial microorganisms [4]. This mucus layer is divided into two layers; intestinal microorganisms primarily exist in the outer layer, with only a small number in the inner layer [5]. In inflammatory bowel disease, cytokines are considered crucial mediators of tissue damage. Studies have shown that tumor necrosis factor-α (TNF-α)-secreting cells increase in inflamed intestinal mucosa, and in patients with ulcerative colitis, interleukin 8 (IL-8) is diffusely expressed throughout the mucosa [6,7]. Lactic acid bacteria (LAB), such as lactobacilli, can adjust the pH to make the environment inhospitable to pathogens, and the toxicity of nosocomial microorganisms may be adjusted by metabolites, such as short-chain fatty acids (SCFAs), generated during the metabolic processes of microorganisms [8,9].

The skin, the body’s largest organ, shares similarities with the intestines in its role as an immune barrier [10,11]. Collagen, a significant component that forms the skin, is primarily composed of amino acids, including glycine (32.9%), proline (12.6%), and hydroxyproline (9.5%) [12]. Similar to how the intestines employ tight junction proteins and mucosal layers to defend against external pathogens and stimuli, collagen is crucial in the skin’s defensive barrier against external stressors. The degradation of collagen fibers, along with other extracellular matrix (ECM) components, is regulated by a class of zinc-dependent endopeptidases known as matrix metalloproteinases (MMPs) [13]. The ECM is composed of fibrous collagen and associated proteins, and cell–ECM interactions in the skin play a crucial role in maintaining homeostasis and influencing aging. These interactions are affected by various external factors, such as ultraviolet (UV) exposure, which contributes to skin photoaging by impacting collagen degradation and altering elastic fiber levels [14,15]. Symbiotic organisms also influence the skin. For instance, symbiotic bacteria like *S. epidermidis* are known to modulate the immune system by inducing the production of various cytokines while suppressing the secretion of TNF-α and interleukin-6 (IL-6) [16,17]. Recent studies have shown that germ-free mice exhibit reduced Toll-like receptor (TLR) expression, antimicrobial peptide (AMP) production, and IL-1 cytokine signaling in the skin compared to mice without specific pathogens [18]. While numerous studies have demonstrated that the ingestion of lactic acid bacteria can mitigate UV-induced skin damage and wrinkle formation, the specific key components contributing to these effects remain unclear [19,20,21].

Extracellular vesicles (EVs) are secreted extracellularly and play a role in conveying information between cells [22]. EVs, which are 20–200 nm in size and surrounded by a lipid bilayer, protect intestinal epithelial cells, and kefir-derived EVs regulate inflammatory responses by alleviating inflammatory cytokine production [23,24]. Recently, exosomes have been used as therapeutic agents to improve intestinal function and related pathological irritable bowel syndrome mechanisms, such as Alzheimer’s disease, Parkinson’s disease, and cardiovascular diseases associated with intestinal dysfunction [25,26]. Although only a few studies have reported the health-promoting effects of exosomes secreted by non-pathogenic microorganisms, including LAB, there is growing recognition of LAB-derived exosomes as promising therapeutic platforms [27,28].

Recent advancements in microfluidic technology have introduced methods to simulate the human gut environment, often called a “gut-on-a-chip” system [29]. This platform features controllable multi-system parameters that allow for the manipulation of microenvironmental conditions [30]. Systems that mimic the human gut’s mechanical, structural, and absorptive properties using living cells in vitro offer potential alternatives to animal testing. Utilizing this system, it is possible to explore the interactions between exosomes and the microfluidic gut-on-a-chip model.

This study aimed to investigate the main functional substances in *Lactiplantibacillus plantarum* HY7714 (HY7714), which was isolated from the breast milk of healthy women and has been clinically confirmed to improve UV-induced skin damage and be safe for consumption without adverse reactions [31]. In addition, we characterized the extracellular vesicles derived from HY7714 and explored their potential effects on the improvement of intestinal inflammation and skin aging. Given the growing interest in the gut–skin axis, we sought to examine further how this axis might mediate the beneficial effects of HY7714 extracellular vesicles. We hypothesized that upon ingestion, specific substances, such as extracellular vesicles secreted by HY7714 in the gut, would contribute to improving both intestinal health and skin aging.

## 2. Materials and Methods

### 2.1. Isolation and Measurement of Bacteria-Derived EVs

*Lpb. plantarum* HY7714 (HY7714) was supplied by hy Co., Ltd. (Yongin-si, Republic of Korea). *L. plantarum* type strain KCTC3108 (KCTC3108) was obtained from the Korean Collection for Type Cultures (KCTC, Jeongeup-si, Republic of Korea). The HY7714 and KCTC3108 strains were grown in 2 L of Modified Strullu and Romand (MSR) medium at 37 °C for 18–24 h, and were pelleted by sequential centrifugation at 15,970× *g* at 10 °C for 15 min. The culture supernatants of the HY7714 and KCTC3108 strains were separated by centrifugation and filtered using a KrosFlo^®^ KR2i TFF System from Repligen (Spectrum Labs, Los Angeles, CA, USA) and a 0.2 μm membrane filter to remove cell residues. *Lpb. plantarum* HY7714- and KCTC3108-derived EVs were obtained using a TFF system 300 kDa membrane filter (Spectrum Labs, Los Angeles, CA, USA). The retentate was concentrated to 20 mL to obtain a 300 kDa-size retentate at a 50-fold concentration. The bacteria-derived EVs were diluted 1:1 with phosphate-buffered saline (PBS) in a 15% maltodextrin solution and dried using a freeze dryer (FDT-8620, Operon, Gimpo, Republic of Korea). The bacteria-derived EVs in powder form were then diluted in PBS to a 50 mg/mL concentration and stored at 4 °C. The particle sizes of the HY7714 and KCTC3108 EVs were determined using a Zetasizer Nano ZS90 (Malvern Instruments Ltd., Malvern, Worcestershire, UK).

### 2.2. Quantification of Bacteria-Derived EVs

The quantification of the isolated exosomes was performed using the EXOCET Exosome Quantitation Assay kit (System Biosciences, Palo Alto, CA, USA), following the manufacturer’s instructions. In brief, 50 µg of each exosome sample was incubated with EXOCET Lysis Buffer at 37 °C for 5 min, then vortexed for 15 s, and centrifuged for 5 min. The samples were subsequently loaded onto a microtiter plate with the reaction buffer. The plate was then incubated at room temperature for 20 min and read using a Gen5 spectrophotometric plate reader (BioTek, Winooski, VT, USA) at a wavelength of 405 nm.

### 2.3. Cryogenic Transmission Electron Microscopy

Quantifoil^®^ R 2/2 (or 1.2/1.3) and 200 (Quantifoil Micro Tools GmbH, Großlöbichau, Germany) EM grids were glow discharged for 60 s at 20 mA and a positive polarity in the air atmosphere (GloQube^®^ Plus, Quorum, Laughton, UK). The Vitrobot conditions were set to 4 °C, 95% relative humidity, blot time: 5 s, and blot force: 4. The sample suspension (4 µL) was applied to the grid, blotted, and plunge-frozen in liquid ethane with the Vitrobot Mark IV (Thermo Fisher Scientific, Waltham, MA, USA). The samples were visualized under cryo-conditions using a Falcon 3EC detector on a 200 kV Glacios microscope (Thermo Fisher Scientific).

### 2.4. Gut-on-a-Chip Technique and Cell Preparation

The gut-on-a-chip was manufactured following a previously described method [30]. The chips were autoclaved at 120 °C for 30 min and coated with 1 mg/mL poly-D-lysine (Sigma-Aldrich, St. Louis, MO, USA) to improve cell adhesion. The chip was washed with deionized water and filled with a clean medium.

Caco-2 human intestinal adenocarcinoma cells were purchased from the Korean Cell Line Bank (Seoul, Republic of Korea). The human intestinal epithelial cells (Caco-2) were cultured in a Dulbecco’s modified Eagle’s medium (GIBCO Invitrogen Life Technologies, Grand Island, NY, USA) with 10% fetal bovine serum (FBS; GIBCO Invitrogen Life Technologies) and penicillin–streptomycin without calcium (GIBCO Invitrogen Life Technologies). The Caco-2 cell suspension was loaded into the microchannel at a 1.0 × 10^6^ cells/mL concentration. The cells in the suspension medium flowed into the microchannels via gravity. After overnight incubation, non-adherent cells were washed with fresh medium and Dulbecco’s phosphate-buffered saline (DPBS). The inlet of the gut chip was connected through a tube to a syringe containing the medium, and the outlet was connected to an empty syringe fixed to a syringe pump (New Era Pump Systems Inc., Farmingdale, NY, USA). The pump was operated at 30 μL/h to create a flow (Figure 1).

### 2.5. Immunofluorescence Staining

After sufficiently culturing the cells, the intestinal cells were weakened by treating them with 15 μg/mL of lipopolysaccharides from *Escherichia coli* O127:B8 (LPS, Sigma-Aldrich). After 24 h of exposure, 1.0 × 10^6^ colony-forming units/mL of bacteria and 10 μg/mL of EVs were added to the cells. EVs derived from two lactic acid bacteria (*Lpb. plantarum* HY7714 and KCTC3108) were used for the treatment. The cells cultured on the gut chip were fixed with 4% paraformaldehyde for 30 min. The fixed cells were washed twice for 5 min with 0.1% bovine serum albumin (BSA; Sigma-Aldrich) dissolved in DPBS and treated with 0.2% Triton X-100 (Sigma-Aldrich) for 30 min. The cells were then incubated with a 2% BSA (*w*/*v*) solution for 1 h. After the cells were cultured with primary antibodies overnight at 4 °C, the cells were washed and incubated with secondary antibodies for 90 min. Three antibody types were used for immuno-histochemistry: rabbit anti-albumin polyclonal antibody (1:500, Invitrogen), Alexa Fluor 594-conjugated phalloidin (1:250, Invitrogen), and donkey anti-rabbit Alexa Fluor 488 (1:1000, Invitrogen). The cells were then incubated with 4′,6-diamidino-2-phenylindole dihydrochloride (Molecular Probes, Eugene, OR, USA), zonula occludens-1 (ZO-1), and rhodamine-phalloidin to visualize the cell nuclei, tight junctions, and F-actin, respectively. Cell images were obtained using a confocal laser-scanning microscope (LSM 700; Carl Zeiss Microscopy, Peabody, MA, USA).

### 2.6. Cell Cultures

Human HT-29 and HS68 (CRL-1635) human dermal fibroblasts were obtained from the American Type Culture Collection (ATCC, Manassas, VA, USA). Human HT-29 cells were grown in MEM containing 10% FBS (GIBCO Invitrogen Life Technologies) and 1% antibiotic–antimycotic (GIBCO Invitrogen Life Technologies). HS68 cells were grown in DMEM with high glucose (GIBCO Invitrogen Life Technologies) containing 10% FBS (GIBCO Invitrogen Life Technologies) and 1% antibiotic–antimycotic (GIBCO Invitrogen Life Technologies). Cultures were incubated at 37 °C in a 5% CO_2_ atmosphere. The culture medium was changed every 2 days. The cells were seeded in 6-well plates (90% confluence) at a density of 1 × 10^6^ cells/mL and starved in a serum-free medium for 24 h before each experiment.

### 2.7. Cell Viability

The cells were cultured in a humidified atmosphere of 5% CO_2_ in an incubator at 37 °C and seeded into 96-well plates at a density of 1 × 10^4^ cells/mL using the cell counting kit-8 (CCK-8) cell viability assay (Dojindo Molecular Technologies, Inc., Kumamoto, Japan). For the HT-29 cells, the cell viability was assessed after treating them with HY7714 EVs for 24 h in the presence of LPS from *Escherichia coli* O127:B8 (2 μg/mL). In the case of HS68 cells, the cell viability was evaluated after 24 h treatment with HY7714 EVs following UVB exposure (30 mJ/cm^2^).

### 2.8. qRT-PCR Analysis

The cells were seeded in 6-well plates at a density of 1 × 10^6^ cells/mL for qRT-PCR. The total RNA was isolated from the cells using an easy-spin total RNA extraction kit (iNtRON Biotechnology, Seongnam, Seoul, Republic of Korea). The extracted total RNA samples were quantified using a NanoDrop 2000 spectrophotometer (Thermo Fisher Scientific, Waltham, MA, USA) and stored at −20 °C until use. cDNA was synthesized using the Maxime RT premix kit (iNtRON Biotechnology). The cDNA samples were amplified using the Quant Studio 6-flex real-time PCR system (Applied Biosystems, Foster City, CA, USA) and gene expression master mix (Applied Biosystems). qRT-PCR was conducted using mouse-specific TaqMan gene expression assays for TNF-ɑ (Hs00174128_m1), IL-8 (Hs00174103_m1), ZO-1 (Hs01551861_m1), occludin (OCLN; Hs00170162_m1), and glyceraldehyde-3-phosphate dehydrogenase (GAPDH; Hs99999905_m1). The mRNA ex-pression level of each gene was calculated using the 2^−ΔΔCt^ method and normalized to that of GAPDH.

### 2.9. Protein Expression Analysis Through Western Blotting

The HT-29 cells were seeded in 6-well culture plates (1 × 10^6^ cells/mL) and incubated in a serum-free medium for 24 h. The cells were lysed on ice in Pro-Prep protein extraction solution containing 1.0 mM phenylmethylsulfonyl fluoride, 1.0 mM EDTA, 1 μM pep-statin A, 1 μM leupeptin, and 0.1 μM aprotinin (iNtRON Biotechnology) after treating LPS-treated HT-29 cells with *Lpb. plantarum* HY7714EVs and KCTC3108EVs and culturing them for 24 h. The cell lysate was harvested by centrifuging at 12,000 rpm for 10 min at 4 °C. The protein extracts (20 μg) were separated on a 4–15% gel (Bio-Rad, Hercules, CA, USA) via sodium dodecyl-sulfate polyacrylamide gel electrophoresis, and transferred to a polyvinylidene difluoride membrane (Thermo Fisher Scientific). The membranes were blocked with 5% skim milk (BD Biosciences, San Jose, CA, USA) in Tris-buffered saline containing 0.1% Tween-20 (TBS-T) for 1 h at room temperature, and incubated with primary antibodies against myeloid differentiation primary response 88 (MyD88), mammalian target of rapamycin (mTOR), phospho-mTOR (p-mTOR), nuclear factor kappa-light-chain-enhancer of activated B cells (NF-κB), phospho-NF-κB (p-NF-κB), and β-actin (1:1000; all purchased from Cell Signaling Technology, Danvers, MA, USA) at 4 °C overnight. The membranes were washed thrice for 5 min each with TBS-T and incubated with a goat anti-rabbit IgG-HRP secondary antibody conjugated with horseradish peroxidase (1:10,000; Cell Signaling Technology) for 60 min at room temperature. The bands were visualized using chemiluminescence (Thermo Fisher Scientific), detected using an Image Quant LAS4000 (GE Healthcare, Madison, WI, USA) imaging system, and quantified using ImageJ software (version 1.47; National Institutes of Health, Bethesda, MD, USA).

### 2.10. Statistical Analyses

All the experimental results are expressed as the mean ± standard error (SE) of independent experiments and were analyzed using GraphPad Prism 5 (GraphPad Software, San Diego, CA, USA). Statistical comparisons between groups were performed using Stu-dent’s *t*-test. A *p* of <0.05 was considered statistically significant.

## 3. Results

### 3.1. Characterizing Lpb. plantarum HY7714 EVs

We investigated the particle-size distribution of the extracellular vesicles (EVs) isolated from the *Lpb. plantarum* HY7714 culture supernatant using a nanoparticle size analyzer with zeta potential (Figure 2A). Since exosomes are spherical particles comprising a phospholipid double membrane, we investigated the shape and size of the particles using transmission electron microscopy. According to cryo-EM analyses of density-purified EVs, EVs have a closed spherical membrane structure (Figure 2B). Morphological assessments revealed a spherical, bilayered, and closed membrane structure with an average diameter of <100 nm [32]. These results indicate that *Lpb. plantarum* spontaneously releases EVs of varying morphology and size. The particles were separated into 24.4–255 nm sizes in HY7714 EVs.

### 3.2. Quantification of HY7714 EVs

We measured the acetyl-CoA acetylcholinesterase activity to quantify the number of extracellular vesicles. The HY7714 had 2.97 × 10^7^ EVs, and the KCTC3108 had 3.77 × 10^7^ EVs (Table 1).

### 3.3. Effect of Intestinal Barrier Enhancement by EVs on Gut-on-a-Chip

The differentiated cells on the chip were treated with the bacteria and EVs, and confocal imaging analysis was performed to evaluate the efficacy of the EVs in treating intestinal cells (Figure 3). After treating the cells with LPS, cell shape destruction and weakening of the cell wall thickness were observed. In addition, tight junctions stained with ZO-1 were rarely observed. Cells treated with the HY7714 EVs showed greater cell thickness and a higher level of tight junctions than the LPS-treated cells and the KCTC3108 EVs-treated group. This finding indicates that the HY7714-derived EVs restored cell function after LPS treatment.

### 3.4. The Cytoprotective and Intestinal Protective Effect of HY7714 EVs on HT-29 Cells

HT-29 cells are mucus-secreting and columnar-absorptive cells [33]. These cells are also used in 3D organotypic models derived from 3D colonic epithelium models because they are involved in microbial pathogenesis [34]. Therefore, HT-29 cells are suitable for elucidating EV–gut interactions in detail. We first performed a cell viability assay to determine whether the HY7714 EVs ameliorated LPS-induced toxicity in HT-29 cells. The viability of the LPS-treated HT-29 cells was 73.60%, significantly lower than that of the control cells (*p* < 0.01). The viability of the LPS-treated HT-29 cells was significantly higher than that of the LPS cells. The data showed that the HY7714 EVs improved cell viability concentration dependently to 81.35% at 25 μg/mL, 82.81% at 50 μg/mL, and 88.95% at 100 μg/mL (Figure 4A).

LPS activates the mTOR and inhibits autophagy. The mTOR-dependent autophagy induces oxidative stress and the production of infectious cytokines. Dysregulation of mTOR phosphorylation and autophagy-related proteins has been identified in inflamed colonic tissues from patients with active ulcerative colitis and is associated with TLR4-MyD88 signaling [35]. We investigated the effects of HY7714 EV treatment on the MyD88/mTOR/NF-κB signaling pathway by examining the expression of MyD88, mTOR, and NF-κB using Western blot analysis. LPS treatment increased protein levels of MyD88 p-mTOR, and p-NF-κB by 171.5% (*p* < 0.05), 191.3% (*p* < 0.05), and 174.7%, respectively. The MyD88, p-mTOR, p-NF-κB protein levels recovered to 110.7%, 162.5%, and 104.5%, respectively, in cells treated with HY7714 EVs (Figure 4B).

LPS secreted by harmful bacteria plays an important pathogenic role in intestinal inflammation and other inflammatory diseases. Increased LPS levels impair intestinal permeability [36,37]. Therefore, we determined the gene expression of TNF-α, IL-8, ZO-1, and OCLN, using quantitative real-time PCR (qRT-PCR). TNF-α and IL-8 mRNA expression levels increased 3.55-fold and 14.70-fold, respectively, compared to the control group (*p* < 0.001). However, when HT-29 cells were treated with 100 μg/mL of HY7714 EVs, TNF-α and IL-8 mRNA levels were significantly reduced to 1.85-fold (*p* < 0.01) and 10.12-fold (*p* < 0.01), compared to the LPS group (Figure 4C,D). The mRNA levels of ZO-1 were 0.55-fold lower in the LPS-treated cells than in the control cells. However, HY7714 EV treatment significantly increased their expression by 0.89-fold (Figure 4E). The OCLN expression level was significantly decreased by 0.60-fold in the LPS cells, but was significantly improved in the HY7714 EVs-treated cells by 1.03-fold (Figure 4F).

### 3.5. RNA-Sequencing Analysis of HY7714 EVs

Through RNA-sequencing analysis of the isolated exosomes, we aimed to gain insights into the gene expression regulation of Lactobacillus exosomes. As a result, loci for a total of 3467 small RNA candidates were identified. Specifically, the loci for each type of small RNA were as follows: 283 for utr5, 2555 for asRNA, and 629 for sRNA. A differential expression analysis of the 3467 small RNA candidates was conducted, revealing 566 differentially expressed genes based on a *p*-value < 0.05 and a Log2 fold change > |2| (Figure 5A). The HeatMap showed distinct gene expression patterns between the EVs of *Lpb. plantarum* KCTC3108 and HY7714, with 44 genes upregulated and 322 genes downregulated by the HY7714 EVs (Figure 5B). Among these, DAVID analysis suggested the potential regulation of two downregulated genes, the vicinal oxygen chelate (VOC) family protein and proline aminopeptidase, involved in protein hydrolysis.

### 3.6. Effect of HY7714 EVs on ECM-Related Gene Expressions in HS68 Cells

Based on the RNA sequencing results, we hypothesized that the downregulation of VOC family proteins and proline aminopeptidase could influence the protection against oxidative stress and collagen structure degradation caused by UVB exposure. First, we investigated the cytoprotective effects of the HY7714 EVs on UVB-induced HS68 cells (Figure 6A). As a result, the survival rate of the UVB-induced HS68 cells was 87.97% (*p* < 0.05). Treatments with TS at 25, 50, and 100 μg/mL resulted in cell survival rates of 75.36%, 84.49%, and 89.31%, respectively. In contrast, with treatments with the HY7714 EVs at concentrations of 25, 50, and 100 μg/mL, the survival rates recovered to 89.85%, 88.88%, and 93.62%, respectively. Subsequently, MMP-1 expression increased by 6.90-fold (*p* < 0.001) due to UVB exposure compared to the control group, but was reduced to 2.09-fold (*p* < 0.001) and 1.46-fold (*p* < 0.01) by the TS EVs and HY7714 EVs, respectively (Figure 6B). A significant difference between the TS EVs and HY7714 EVs was also observed (*p* < 0.001). MMP-3 expression increased by 10.71-fold (*p* < 0.001) due to UVB exposure compared to the control group, but was reduced to 2.36-fold (*p* < 0.001) and 1.77-fold (*p* < 0.001) by the TS EVs and HY7714 EVs, respectively (Figure 6C). A significant difference between the TS EVs and HY7714 EVs was also observed (*p* < 0.01). Serine palmitoyl transferase (SPT)-1 expression decreased to 0.07-fold (*p* < 0.001) due to UVB exposure compared to the control group, but was improved to 0.96-fold and 0.82-fold by the TS EVs and HY7714 EVs, respectively (Figure 6D). There was no significant difference between the TS EVs and HY7714 EVs.

## 4. Discussion

The human gut contains a vast and diverse microbial ecosystem essential for human health [38]. This microbial ecosystem protects the host from pathogens by promoting host defense mechanisms [39]. In this respect, the usefulness of various strains of lactic acid bacteria have been reported [40]. Although the pathogenesis of inflammatory bowel disease (IBD) has not yet been clearly elucidated, various environmental factors are caused by passing immune responses in the intestinal mucosa [41]. Adhesive *E. coli* is abundant in the stool of patients with ulcerative colitis [42]. These pathogens can deposit on the surface of the mucosa and invade the inside of the mucosa, so it is essential to distinguish them and control the immune response [43,44].

Recent work carried out on the ability of food-derived nanoparticles to modulate the expression of intestinal carrier genes or immune-related genes has found them to have protective effects on intestinal cells, and these nanoparticles have been proposed as new functional substances [45,46]. Among nanoparticles, exosomes mediate intercellular communication and cause changes in aspects such as gene and protein expression [47]. However, research has not fully understood whether lactic acid bacteria-derived exosomes improve the cell wall, and can prevent intestinal leakage caused by inflammation in intestinal cells, as well as alleviating the associated inflammatory factors. EVs secreted by *Lpb. plantarum* HY7714, which colonizes the outer mucus layer, can interact with the intestines through the inner mucus layer, potentially regulating the number of microorganisms that threaten human health. To address the hypothesis that these EVs could have beneficial effects on human health, we conducted the following study.

First, we evaluated the barrier-protective effects of extracellular vesicles using a lipid polysaccharide-induced intestinal injury and inflammation model, gut-on-a-chip. To determine whether HY7714 EVs affected this barrier, we compared the intestinal cell wall thickness of the LPS-treated and HY7714 EV-treated groups using a gut-on-a-chip. It was found that HY7714-derived EVs restored cellular functions weakened by LPS treatment. In intestinal tissues exposed to the external environment, the intestinal barrier collapses and the intestinal mucosa is exposed to bacteria, causing an inflammatory response [48]. Therefore, we confirmed that HY7714-derived EVs effectively prevented membrane permeability, improved barrier function, and protected intestinal tissue.

We then evaluated EV–intestinal interactions using HT-29 cells, which contain mucus-secreting cells and columnar absorptive cells, and are also used as a model for microbial pathogenesis. Consequently, HY7714 EVs can protect against an LPS-induced cell viability decrease in HT-29 cells. In greater detail, we investigated how HY7714-derived EVs enhance intestinal function and contribute to the regulation of intestinal inflammation.

Cytokines and oxidative stress in the gastrointestinal tract modulate inflammatory processes in various intestinal diseases [49,50]. MyD88, an adapter protein, mediates the signaling of most TLRs and induces pro-inflammatory cytokine production [51]. The role of auto-predation in regulating inflammation is also crucial, with recent studies linking the TLR-MyD88 and mTOR pathways to intestinal mucosal destruction and attenuation of ulcerative colitis development [35,52]. Pro-inflammatory cytokine levels are reduced after LPS treatment of mTOR-silenced cells [35]. We investigated how HY7714 EVs influence the MyD88/mTOR/NF-κB signaling pathway and found that treatment with HY7714 EVs improved the expression of MyD88, mTOR, and NF-κB proteins in HT-29 cells activated by LPS. The pro-inflammatory cytokines TNF-α and IL-8 were also reduced to control levels following HY7714 EV treatment. In addition, HY7714 EVs increase the mRNA expression levels of ZO-1 and OCLN, which are related to tight junctions. These findings suggest that HY7714 EVs may help to alleviate intestinal inflammation by modulating pro-inflammatory cytokines through the MyD88/mTOR/NF-κB pathway, showing results consistent with previous studies on EVs derived from other lactic acid bacteria [53,54,55]. Furthermore, these results indicate that EVs secreted by lactic acid bacteria may mediate intercellular signaling within the intestine, contributing to inflammation regulation and the improvement of tight junction integrity.

The previous studies we referenced primarily focused on evaluating the efficacy of lactic acid bacteria-derived EVs in alleviating intestinal inflammation. In this study, we performed miRNA sequencing of HY7714-derived EVs to compare their RNA components with those of the standard strain, aiming to predict additional functionalities. As a result, we observed the downregulation of VOC family proteins and proline aminopeptidase, which are associated with the regulation of protein hydrolysis. Reducing VOC family proteins may alleviate oxidative stress and metal-ion-catalyzed reactions [56], thereby mitigating cellular aging and extracellular matrix degradation, which could have a beneficial effect on collagen preservation and the skin’s aging process. The downregulation of these proteins may help prevent the excessive accumulation of reactive oxygen species (ROS) or reactive intermediates, thereby maintaining collagen integrity and reducing signs of aging. Similarly, reducing oxidative stress through the regulation of VOC proteins may have a significant impact on gut health. Oxidative stress is known to damage the intestinal epithelial barrier, which can result in increased intestinal permeability (“leaky gut”) and subsequent systemic inflammation [57]. By controlling VOC proteins and limiting oxidative stress, HY7714 EVs could help to maintain intestinal barrier integrity, thereby reducing leaky gut and the levels of inflammation markers. In the gut, the degradation of collagen or other structural proteins of the intestinal lining can weaken barrier function [58]. Thus, the inhibition of this enzyme may not only prevent collagen degradation in the skin, but also help to maintain the structural integrity of the gut lining, contributing to overall gut health and reduced inflammation.

This hypothesis is based on the premise that VOC family proteins may promote oxidative reactions or contribute to the degradation of structural proteins, such as collagen, through their catalytic activities in various chemical reactions. Therefore, reducing VOC proteins may positively inhibit collagen degradation and prevent skin aging, particularly under conditions where oxidative stress plays a significant role. Proline aminopeptidase plays a role in releasing N-terminal proline from peptides [59]. Since collagen is rich in proline and hydroxyproline, excessive activation of proline aminopeptidase may destabilize the collagen structure. Based on this, we hypothesized that inhibiting this enzyme could help to prevent collagen degradation and contribute to the maintenance of skin elasticity. We aimed to investigate the anti-aging effects of HY7714 EVs on skin cells and found that the EVs improved cell viability, which had been impaired by UVB exposure. Furthermore, we observed that the EVs ameliorated the UVB-induced upregulation of MMP-1, MMP-3, and SPT-1 expression, with the improvement of MMP-1 and MMP-3 being more pronounced in the HY7714 EVs than in the TS EVs. Given the shared mechanisms of collagen preservation and barrier integrity in both the skin and gut, it is plausible that the beneficial effects of HY7714 EVs on collagen preservation in the skin could extend to maintaining the structural integrity of the gut lining. By mitigating oxidative stress and reducing proteolytic enzyme activity, HY7714 EVs may improve skin anti-aging and gut health. The previously mentioned studies focused on describing the singular effects of lactic acid bacteria-derived EVs. In contrast, our findings highlight the dual effects of these EVs from the perspective of the gut–skin axis.

In conclusion, the findings suggest that HY7714 EVs play a dual role in enhancing skin and gut health by modulating oxidative stress, reducing enzymatic degradation of structural proteins, and maintaining barrier integrity. In light of this study, considering the shared mechanisms of collagen preservation and barrier integrity in both the skin and intestines, it is plausible that the beneficial effects of HY7714 EVs on collagen preservation in the skin may also extend to maintaining the structural integrity of the intestinal lining. HY7714 EVs are thought to prevent skin aging and promote intestinal health by alleviating oxidative stress and reducing proteolytic enzyme activity. Future studies should focus on further elucidating the interconnected pathways between skin and gut health, emphasizing how EVs like HY7714 could provide holistic anti-aging benefits. Moreover, our research included not only genetic material but also intracellular materials, such as unnecessary intracellular proteins and materials necessary for targeting, making it challenging to confirm changes when administered in the body. Therefore, a clinical study will be required. In addition, this study did not explain all the mechanisms related to the protective effect of HY7714 EVs on the intestines; detailed studies must be conducted to elucidate the mechanisms of HY7714 EVs. Furthermore, when ingesting extracellular vesicles, their composition and morphology may change because of gastric acids or enzymes; therefore, acid and bile resistance tests should be conducted to confirm that they are unaffected.

## 5. Conclusions

The results of this study demonstrate that EVs secreted by LAB interact with cells to improve markers related to intestinal inflammation. We confirmed that EVs derived from HY7714 improve intestinal health by mitigating gut inflammation and leaky gut, and that they alleviate UVB-induced oxidative stress through skin–gut communication. Therefore, HY7714 EVs may be valuable and safe therapeutic agents for improving intestinal health, including barrier strengthening. Furthermore, HY7714 EVs may help to prevent skin photoaging by reducing oxidative damage and supporting the skin’s resilience against UVB-induced aging processes. However, further research is needed to determine their detailed gut and skin health mechanisms.

## Figures and Tables

**Figure 1 microorganisms-12-02466-f001:**
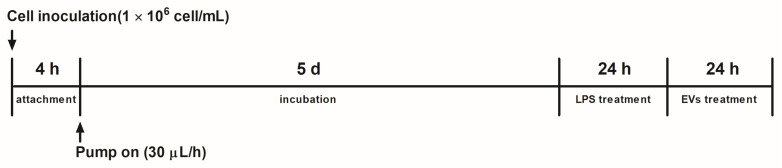
The flowchart of the gut-on-a-chip.

**Figure 2 microorganisms-12-02466-f002:**
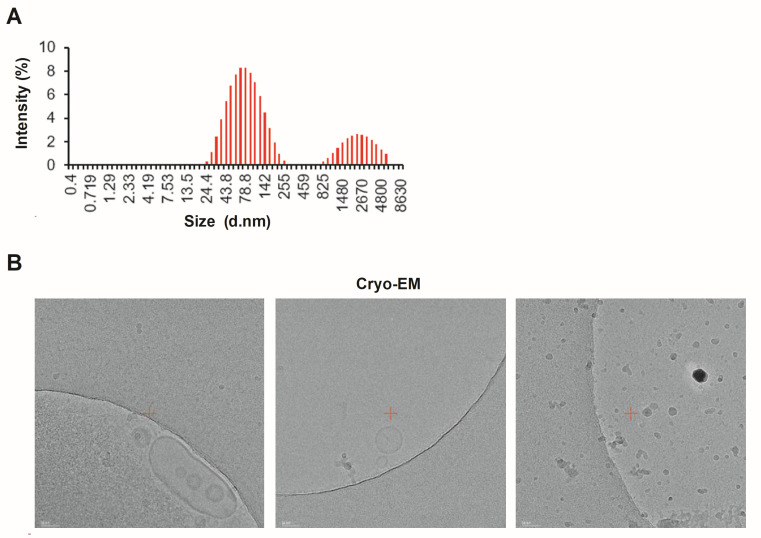
Characterization of extracellular vesicles (EVs). (**A**) Size distribution determined by intensity of *Lpb. plantarum* HY7714 EVs. (**B**) Cryo-EM images of HY7714 EVs. Cryo-EM image analyses of density-purified HY7714 EVs. The out-lined EV images are enlarged, and indicate the lipid bilayer. Scale bars, 50 nm. Red focus sign: Focusing of center is needed to stabilize the stage before recording are performed.

**Figure 3 microorganisms-12-02466-f003:**
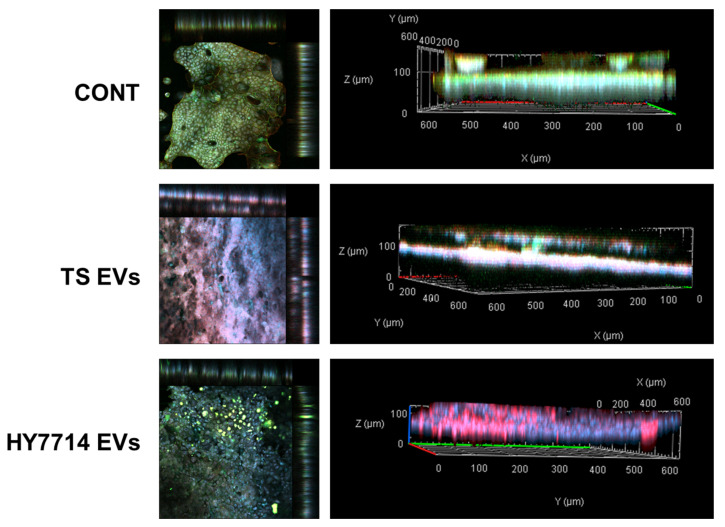
Evaluation of intestinal barrier function of EVs applied to the gut-on-a-chip. Immunostaining of Caco-2 cells on the gut chip. The nuclei were stained blue with 4′,6-diamidino-2-phenylindole dihydrochloride. Tight junction and F-actin were colored green and red due to zonula occludens-1 (ZO-1) and rhodamine-phalloidin. CONT, control; TS, *Lpb. plantarum* type strain KCTC3108; TS EVs, TS extracellular vesicles; HY7714, *Lpb. plantarum* HY7714; HY7714EVs, HY7714 extracellular vesicles.

**Figure 4 microorganisms-12-02466-f004:**
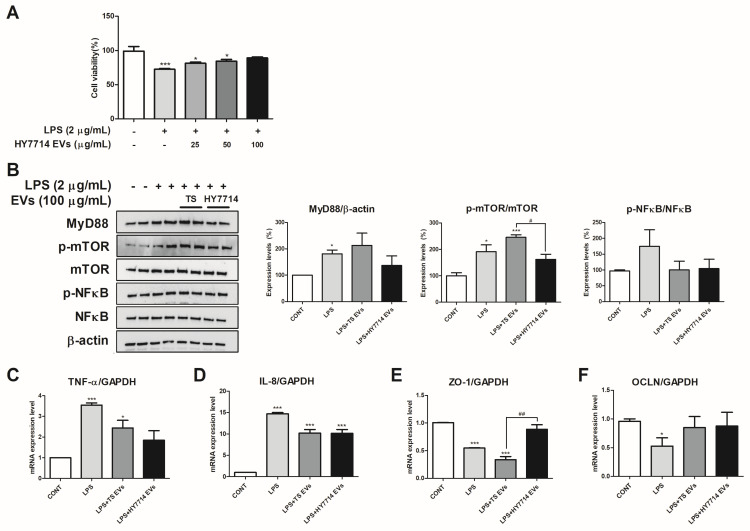
Protective effects and improvement of intestinal inflammation by EVs on HT-29 cells. (**A**) The protective effect of *Lpb. plantarum* HY7714 EVs in LPS-induced HT-29 cells. (**B**) The protein levels of myeloid differentiation primary response 88 (MyD88), phospho-mammalian target of rapamycin (p-mTOR), mTOR, phospho-nuclear factor kappa-light-chain-enhancer of activated B cells (p-NF-κB), NF-κB, and β-actin. The mRNA levels of (**C**) tumor necrosis factor-α (TNF-α), (**D**) interleukin 8 (IL-8), (**E**) zonula occludens-1 (ZO-1), and (**F**) occludin (OCLN) were monitored using qPCR and normalized against glyceraldehyde 3-phosphate dehydrogenase (GAPDH). The data are presented as the mean ± SE. Significant differences are indicated by * *p* < 0.05 and *** *p* < 0.001 relative to the control group. Significant differences are indicated by # *p* < 0.05 and ## *p* < 0.01 relative to the TS group. SE, standard error; CONT, control; EVs, extracellular vesicles; TS EVs, *Lpb. plantarum* type strain KCTC3180 EVs; HY7714 EVs, *Lpb. plantarum* HY7714 EVs.

**Figure 5 microorganisms-12-02466-f005:**
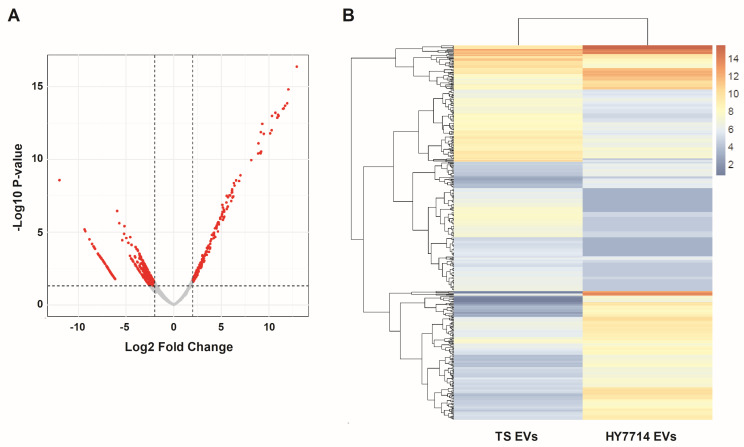
Differential expression analysis of 3467 small RNAs from exosomes of *Lactiplantibacillus plantarum* KCTC3108 and HY7714. (**A**) Volcano plot showing differentially expressed genes identified using the criteria of *p*-value < 0.05 and Log2 Fold Change > |2| (red dots). (**B**) Heatmap displaying 44 upregulated and 322 downregulated genes in HY7714-derived EVs. EVs, extracellular vesicles; TS EVs, *Lpb. plantarum* type strain KCTC3180 EVs; HY7714 EVs, *Lpb. plantarum* HY7714 EVs.

**Figure 6 microorganisms-12-02466-f006:**
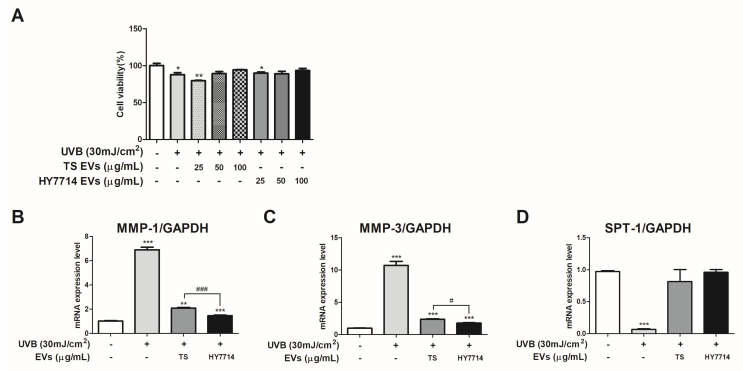
Protective effect of *Lpb. plantarum* HY7714 EVs on HS68 cells. (**A**) Effect of different concentrations (25, 50, and 100 μg/mL) of HY7714 EVs on viability. (**B**–**D**) Effect of HY7714 EVs on ECM-related gene expressions. Data are presented as the mean ± SE. Significant differences are indicated by * *p* < 0.05, ** *p* < 0.01, and *** *p* < 0.001 relative to the control group. Significant differences are indicated by # *p* < 0.05 and ### *p* < 0.001 relative to the TS group. SE, standard error; CONT, control; EVs, extracellular vesicles; TS EVs, *Lpb. plantarum* type strain KCTC3180 EVs; HY7714 EVs, *Lpb. plantarum* HY7714 EVs.

**Table 1 microorganisms-12-02466-t001:** Extracellular vesicle (EV) quantification assay of *Lpb. plantarum* extracellular vesicles (EVs).

Samples	^1^ Quantification of EVs
*Lpb. plantarum* HY7714 EVs	2.97 × 10^7^
*Lpb. plantarum* KCTC3108 EVs	3.77 × 10^7^

^1^ The data represent the average value of three repetitions (optical density of 405 nm).

## Data Availability

The original contributions presented in the study are included in the article, further inquiries can be directed to the corresponding author.

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
