# Peer review of "Targeting Inflammation and Skin Aging via the Gut–Skin Axis: The Role of Lactiplantibacillus plantarum HY7714-Derived Extracellular Vesicles"

_microorganisms, 2024, doi:10.3390/microorganisms12122466_

Round 1
Reviewer 1 Report
Comments and Suggestions for Authors
microorganisms-3303308-peer-review-v1
The paper is interesting and deserve attention of the Editor. However, the way of presenting deserve extensive revision.
Ln46: In accordance with changes in taxonomy of former genus Lactobacillus in this context was recommend using an English word, "lactobacilli", written with not capital L and not italics.
Ln52: what is % of proline?
Ln67: Please, explain meaning of EV here, and not on next line.
Introduction is complete and provide sufficient information on explored subjects, however, it is missing link between different parts. In current way, it is just collection of different, not organically link parts. Please, try to correct slightly and add links between different parts of the introduction.
Ln85: If you have stated that Lactiplantibacillus plantarum HY7714 is a probiotic, then, provide reference where these properties were described and evaluated. Moreover, to be call probiotic, an appropriate animal model and human trails need to be confirming that. In other case, strain need to be call "potential probiotic". Please, correct.
Please, for the abbreviation of former Lactobacillus, use the recommendations https://doi.org/10.1163/18762891-20230114
Authors need to provide details on origin of studied Lpb. plantarum HY7714 strain and reference where isolation, identification, safety and beneficial properties were described.
Ln97: what medium?
Ln 108-112: Please, provide more details
Ln 115-121: Please, provide more details.
Ln124: How can be overnight for 30 min? Please, check entire manuscript and maybe help from professional linguist and/or more experience collogues will be a good option.
Ln143: what kind of LPC? Please, be more specific.
Material and methods need to be described with sufficient details. In high extend paper need this adjustment.
Is there any scientific proof that strain KCTC3108 is a probiotic? provide reference.
Ln171: CO2, 2 needs to be in index position.
Ln172: is the cells concentration correct? log14?
Results are presented well, but reading the manuscript, giving again impression that not all details of the applied methodology were well presented in the Material and methods section.
Discussion need to be improved. In several parts authors give statements that need to be confirmed with appropriate references, and that references are missing. Please, pay attention to the formatting the manuscript. In some cases, is missing italics (example Ln 356). Some of the statement are quite general and not really needed.
Discussion is a bit general. All the interesting results obtained by authors merit a better discussion section, where all obtained results can be discussed appropriately, compared with other study or build hypothesis regarding positive (or negative) effects of the observed facts. Again, maybe help from more experience colleges can be a good option to improve discussion part and paper in general.
Comments on the Quality of English Languagemicroorganisms-3303308-peer-review-v1
The paper is interesting and deserve attention of the Editor. However, the way of presenting deserve extensive revision.
Ln46: In accordance with changes in taxonomy of former genus Lactobacillus in this context was recommend using an English word, "lactobacilli", written with not capital L and not italics.
Ln52: what is % of proline?
Ln67: Please, explain meaning of EV here, and not on next line.
Introduction is complete and provide sufficient information on explored subjects, however, it is missing link between different parts. In current way, it is just collection of different, not organically link parts. Please, try to correct slightly and add links between different parts of the introduction.
Ln85: If you have stated that Lactiplantibacillus plantarum HY7714 is a probiotic, then, provide reference where these properties were described and evaluated. Moreover, to be call probiotic, an appropriate animal model and human trails need to be confirming that. In other case, strain need to be call "potential probiotic". Please, correct.
Please, for the abbreviation of former Lactobacillus, use the recommendations https://doi.org/10.1163/18762891-20230114
Authors need to provide details on origin of studied Lpb. plantarum HY7714 strain and reference where isolation, identification, safety and beneficial properties were described.
Ln97: what medium?
Ln 108-112: Please, provide more details
Ln 115-121: Please, provide more details.
Ln124: How can be overnight for 30 min? Please, check entire manuscript and maybe help from professional linguist and/or more experience collogues will be a good option.
Ln143: what kind of LPC? Please, be more specific.
Material and methods need to be described with sufficient details. In high extend paper need this adjustment.
Is there any scientific proof that strain KCTC3108 is a probiotic? provide reference.
Ln171: CO2, 2 needs to be in index position.
Ln172: is the cells concentration correct? log14?
Results are presented well, but reading the manuscript, giving again impression that not all details of the applied methodology were well presented in the Material and methods section.
Discussion need to be improved. In several parts authors give statements that need to be confirmed with appropriate references, and that references are missing. Please, pay attention to the formatting the manuscript. In some cases, is missing italics (example Ln 356). Some of the statement are quite general and not really needed.
Discussion is a bit general. All the interesting results obtained by authors merit a better discussion section, where all obtained results can be discussed appropriately, compared with other study or build hypothesis regarding positive (or negative) effects of the observed facts. Again, maybe help from more experience colleges can be a good option to improve discussion part and paper in general.
Author Response
Dear reviewer 1
We sincerely thank you for reviewing our manuscript for publication in microorganisms. We are truly honored to have the opportunity to revise our work and deeply appreciate the insightful comments you provided. Your thoughtful suggestions have greatly contributed to improving the quality of our manuscript. We sincerely appreciate all valuable comments and suggestions. We hope that the manuscript is now acceptable for publication in microorganisms and declare that authors of this work have no conflict of interests.
Sincerely,
Soo-Dong Park, Ph.D
Response to Reviewer 1 Comments
Ln46: In accordance with changes in taxonomy of former genus Lactobacillus in this context was recommend using an English word, "lactobacilli", written with not capital L and not italics.
We have revised the content in accordance with your feedback.
-Answer: We have revised the content in accordance with your feedback. (Line 48)
Ln52: what is % of proline?
-Answer: In our initial citation, we referenced a study that summarized the amino acid composition of collagen as the combined percentage of proline and hydroxyproline (22%). However, a more precise analysis of collagen's amino acid composition indicates that the proportions are 32.9% glycine, 12.6% proline, and 9.5% hydroxyproline. These details are documented in the following source: Paul Szpak, "Fish bone chemistry and ultrastructure: implications for taphonomy and stable isotope analysis," Journal of Archaeological Science, Volume 38, Issue 12, 2011, Pages 3358-3372.
Based on your valuable advice, we have revised the content to accurately describe the individual proportions of the three amino acids. (Line 52-55)
Ln67: Please, explain meaning of EV here, and not on next line.
-Answer: We sincerely thank you for pointing out our mistake. We have included an explanation regarding EVs in the manuscript (Line 87).
Introduction is complete and provide sufficient information on explored subjects, however, it is missing link between different parts. In current way, it is just a collection of different, not organically linked parts. Please, try to correct slightly and add links between different parts of the introduction.
-Answer: We sincerely appreciate comments, which helped us to improve the quality of the article. Reflecting upon your valuable feedback, we have endeavored to enhance the coherence between the paragraphs in the introduction section. We kindly request you to review it again to ensure it has been improved. (Line 55-64, 73-76, 83-86)
Ln85: If you have stated that Lactiplantibacillus plantarum HY7714 is a probiotic, then, provide reference where these properties were described and evaluated. Moreover, to be call probiotic, an appropriate animal model and human trails need to be confirming that. In other case, strain need to be call "potential probiotic". Please, correct.
-Answer: Thank you for your kindly advice. Reflecting your valuable feedback, we have excluded the term "Probiotic". HY7714 has undergone animal and clinical trials related to skin health, and studies have been conducted on its effects on microbiome alterations. However, we have determined that the evaluation of the intrinsic efficacy of the probiotic itself remains insufficient. (Line 108-110)
Please, for the abbreviation of former Lactobacillus, use the recommendations https://doi.org/10.1163/18762891-20230114 Authors need to provide details on origin of studied Lpb. plantarum HY7714 strain and reference where isolation, identification, safety and beneficial properties were described.
-Answer: Based on the referenced studies you kindly provided, I have respectfully revised the abbreviation for the genus. Additionally, based on your invaluable feedback, we have incorporated the origin of the strain and other content into the final part of the introduction, along with the relevant references. (Line 121, 128, 179, 231, 254, 256, 262, 266, 273-274, 288-289, 326, 334-335, 343, 353, 375, 380-381, 388, 405)
Ln97: what medium?
-Answer: We sincerely apologize for the oversight and have now respectfully included the name of the medium that was previously not described. (Line 103)
Ln 108-112: Please, provide more details Ln 115-121: Please, provide more details.
-Answer: Based on your valuable feedback, we have further refined the method section to include more specific details. (Line 120-127)
Ln124: How can be overnight for 30 min? Please, check entire manuscript and maybe help from professional linguist and/or more experience colllogues will be a good option.
-Answer: We deeply apologize for our oversight. We have removed the term "overnight" and reflected this correction in the manuscript (Line 130).
Ln143: what kind of LPC? Please, be more specific.
-Answer: Thank you for your kind and thoughtful comments. The LPS used in this study is Lipopolysaccharides from Escherichia coli O127:B8, and we have added this information to the manuscript (Line 210).
Materials and methods need to be described with sufficient details. In high extend paper need this adjustment.
-Answer: We sincerely appreciate your comments, which have greatly helped improve the quality of our manuscript. We have revised the section to provide a clearer explanation.
Is there any scientific proof that strain KCTC3108 is a probiotic? provide reference.
-Answer: Thank you for your kind comments. Based on your suggestion, we have revised the manuscript to replace the term "probiotic" with "lactic acid bacteria (LAB)" (Lines 13, 48, 98, 107, 121, 180-181, 487).
Ln171: CO2, 2 needs to be in index position.
-Answer: We apologize for our mistake. The issue has been corrected to "CO2" in the manuscript (Lines 201, 206).
Ln172: is the cells concentration correct? log14?
-Answer: Thank you for your comments. The cell density used in the study is 1 x 10⁴ cells/mL.
Results are presented well, but reading the manuscript, giving again impression that not all details of the applied methodology were well presented in the Material and methods section.
Discussion needs to be improved. In several parts authors give statements that need to be confirmed with appropriate references, and that references are missing. Please, pay attention to the formatting of the manuscript. In some cases, is missing italics (example Ln 356). Some of the statements are quite general and not really needed.
Discussion is a bit general. All the interesting results obtained by authors merit a better discussion section, where all obtained results can be discussed appropriately, compared with other study or build hypothesis regarding positive (or negative) effects of the observed facts. Again, maybe help from more experience colleges can be a good option to improve discussion part and paper in general.
-Answer: We sincerely appreciate your kind and thoughtful suggestions, which have significantly contributed to improving the quality of our manuscript. Following your advice, we have added appropriate references to the discussion section and corrected the oversight of including general statements without italicization. Furthermore, we have revised the discussion section to ensure a more robust and coherent argument. We truly grateful for your invaluable guidance.

Reviewer 2 Report
Comments and Suggestions for Authors
Thank you to the authors for the opportunity to read the manuscript entitled:
“Targeting Inflammation and Skin Aging via Gut-Skin Axis: 2 Role of Lactiplantibacillus plantarum HY7714-Derived 3 Extra-cellular Vesicles”
I find the article very valuable as it brings new knowledge to explain the multidirectional positive effects of probiotic bacteria on the gut and protecting the skin against damaging environmental factors. I have no comments on the experimental design and the chosen methodology. In my opinion, the conclusions are supported by the reliably obtained results.
The article is very carefully written on the editorial side; however, I “managed to find” the imperfections listed below, which should be corrected before accepting the manuscript for publication.
Line 19: I would ask the authors to consider whether, alongside the abbreviation VOC, an expansion of this name should not appear in this line. Which has only been done in line 308?
Lines 36-38 : „These invasive pathogens penetrate mucous mem‐36 branes and destroy the epithelial barrier, mucus, and red and white blood cells in the stool, 37 causing an acute inflammatory response [2,3].” – I ask the authors to ascertain whether the "red and white blood cells" refer to their appearance in the stool or to their destruction in the stool by pathogens? I currently find this sentence ambiguous.
Line 147; „Sigma” should be?
Line 175: for clarity, I suggest adding the explanation of LPS abbreviation
Line 218: the title „Characterizing L. plantarum HY7714 „ suggests that it is about the characterisation of the strain, whereas from the text and the objective of the work it was rather about the characterisation of the EVs secreted by this bacterium. Please consider changing this title for clarity, e.g., by adding EVs at the end of the current title ?
Line 349: “In this respect, the usefulness of each strain of lactic acid bacteria has been reported [41].” - I leave it to the authors to decide, but in my opinion, the word ‘each’ in this sentence is “a very strong” and responsible statement, suggesting that the authors of the quoted article have confirmed the probiotic characteristics in each strain of lactic bacteria
Author Response
Dear reviewer 2
Thank you for reviewing our manuscript for publication on microorganisms. We are very pleasured to have the opportunity to revise our manuscripts, and appreciate for your insights. As the reviewer noted, we have carefully addressed the reviewer’s insightful feedback, including clarifying ambiguous statements, refining abbreviations, and aligning the title with the manuscript's focus. Your detailed comments greatly enhanced the clarity and quality of our work, and we sincerely appreciate your valuable suggestions. We hope that the manuscript is now acceptable for publication in microorganisms and declare that authors of this work have no conflict of interests.
Sincerely,
Soo-Dong Park, Ph.D
Response to Reviewer 2 Comments
Line 19: I would ask the authors to consider whether, alongside the abbreviation VOC, an expansion of this name should not appear in this line. Which has only been done in line 308?
-Answer: We appreciate for pointing out our mistake. We modified some words, as your advice (Line 19-20)
Lines 36-38 : „These invasive pathogens penetrate mucous mem‐36 branes and destroy the epithelial barrier, mucus, and red and white blood cells in the stool, 37 causing an acute inflammatory response [2,3].” – I ask the authors to ascertain whether the "red and white blood cells" refer to their appearance in the stool or to their destruction in the stool by pathogens? I currently find this sentence ambiguous.
-Answer: Thank you for your thoughtful comment and for pointing out the potential ambiguity in our statement. After reviewing the text, we confirm that the reference to “red and white blood cells” pertains to their presence in the stool as a result of the disruption caused by invasive pathogens, rather than their direct destruction by the pathogens.
To improve clarity, we have revised the sentence as follows:
“These invasive pathogens penetrate mucous membranes and disrupt the epithelial barrier and mucus, leading to the presence of red and white blood cells in the stool and causing an acute inflammatory response [2,3].”
We appreciate your attention to detail and hope this revision resolves any confusion. Thank you again for your valuable feedback.
Line 147; „Sigma” should be?
- Answer: Thank you very much for pointing out this mistake. Following your valuable suggestion, we have accurately revised the term.
Line 175: for clarity, I suggest adding the explanation of LPS abbreviation
-Answer: We deeply appreciate your attention to detail. We have expanded the abbreviation as suggested.
Line 218: the title „Characterizing L. plantarum HY7714 „ suggests that it is about the characterisation of the strain, whereas from the text and the objective of the work it was rather about the characterisation of the EVs secreted by this bacterium. Please consider changing this title for clarity, e.g., by adding EVs at the end of the current title ?
-Answer: Thank you for your excellent suggestion. We have revised the title by adding the term "EVs" for better clarity.
Line 349: “In this respect, the usefulness of each strain of lactic acid bacteria has been reported [41].” - I leave it to the authors to decide, but in my opinion, the word ‘each’ in this sentence is “a very strong” and responsible statement, suggesting that the authors of the quoted article have confirmed the probiotic characteristics in each strain of lactic bacteria
- Answer: Thank you for pointing this out. We agree with your observation that the word “each” may convey a stronger implication than intended. To ensure clarity and accuracy, we have revised the sentence as follows:
“In this respect, the usefulness of various strains of lactic acid bacteria has been reported [41].”
We appreciate your valuable feedback and have made this change accordingly.

Round 2
Reviewer 1 Report
Comments and Suggestions for Authors
Authors have corrected the manuscript and in present way text was improved